# MULTI-TASK OPTION LEARNING AND DISCOVERY FOR STOCHASTIC PATH PLANNING

## ABSTRACT

This paper addresses the problem of reliably and efficiently solving broad classes of long-horizon stochastic path planning problems. Starting with a vanilla RL formulation with a stochastic dynamics simulator and an occupancy matrix of the environment, our approach computes useful options with policies as well as high-level paths that compose the discovered options.

Our main contributions are (1) data-driven methods for creating abstract states that serve as end points for helpful options, (2) methods for computing option policies using auto-generated *option guides* in the form of dense pseudo-reward functions, and (3) an overarching algorithm for composing the computed options. We show that this approach yields strong guarantees of executability and solvability: under fairly general conditions, the computed option guides lead to composable option policies and consequently ensure downward refinability. Empirical evaluation on a range of robots, environments, and tasks shows that this approach effectively transfers knowledge across related tasks and that it outperforms existing approaches by a significant margin.

## 1 INTRODUCTION

Autonomous robots must compute long-horizon motion plans (or path plans) to accomplish their tasks. Robots use controllers to execute these motion plans by reaching each point in the motion plan. However, the physical dynamics can be noisy and controllers are not always able to achieve precise trajectory targets. This prevents robots from deterministically reaching a goal while executing the computed motion plan and increases the complexity of the motion planning problem. Several approaches (Schaul et al., 2015; Pong et al., 2018) have used reinforcement learning (RL) to solve multi-goal stochastic path planning problems by learning goal-conditioned reactive policies. However, these approaches work only for short-horizon problems (Eysenbach et al., 2019). On the other hand, multiple approaches have been designed for handling stochasticity in motion planning (Alterovitz et al., 2007; Sun et al., 2016), but they require discrete actions for the robot. This paper addresses the following question: Can we develop effective approaches that can efficiently compute plans for long-horizon continuous stochastic path planning problems? In this paper, we show that we can develop such an approach by learning abstract states and then learning options that serve as actions between these abstract states.

Abstractions play an important role in long-horizon planning. Temporally abstracted high-level actions reduce the horizon of the problem in order to reduce the complexity of the overall decision-making problem. E.g., a task of reaching a location in a building can be solved using abstract actions such as "go from room A to corridor A", "reach elevator from corridor A", etc., if one can automatically identify these regions of saliency. Each of these actions is a temporally abstracted action. Not only do these actions reduce the complexity of the problem, but they also allow the transfer of knowledge across multiple tasks. E.g, if we learn how to reach room B from room A for a task, we can reuse the same solution when this abstract action is required to solve some other task.

Reinforcement learning allows learning policies that account for the stochasticity of the environment. Recent work (Lyu et al., 2019; Yang et al., 2018; Kokel et al., 2021) has shown that combining RL with abstractions and symbolic planning has enabled robots to solve long-horizon problems that require complex reasoning. However, these approaches require hand-coded abstractions. In this paper,

we show that the abstractions automatically learned (Shah & Srivastava, 2022) can be efficiently combined with deep reinforcement learning approaches.

The main contributions to this paper are: (1) A formal foundation for constructing a library of two different types of options that are task-independent and transferable, (2) A novel approach for auto-generating dense pseudo-reward function in the form of *option guides* that can be used to learn policies for synthesized options, and (3) An overall hierarchical algorithm approach that uses combines these automatically synthesized abstract actions with reinforcement learning and uses them for multi-task long-horizon continuous stochastic path planning problems. We also show that these options are composable and can be used as abstract actions with high-level search algorithms.

Our formal approach provides theoretical guarantees about the composability of the options and their executability using an option guide. We present an extensive evaluation of our approach using two separates sets of automatically synthesized options in a total of 14 settings to answer three critical questions: (1) Does this approach learn useful high-level planning representations? (2) Do these learned representations support transferring learning to new tasks?

The rest of the paper is organized as follows: Sec. 2 some of the existing approaches that are closely related to our approach; Sec. 3 introduces a few existing ideas used by our approach; Sec. 4 presents our algorithm; Sec. 5 presents an extensive empirical evaluation of our approach.

## 2 RELATED WORK

To the best of our knowledge, this is the first approach that uses a data-driven approach for synthesizing transferable and composable options and leverages these options with a hierarchical algorithm to compute solutions for stochastic path planning problems It builds upon the concepts of abstraction, stochastic motion planning, option discovery, and hierarchical reinforcement learning and combines reinforcement learning with planning. Here, we discuss related work from each of these areas.

Motion planning is a well-researched area. Numerous approaches( (Kavraki et al., 1996; LaValle, 1998; Kuffner & LaValle, 2000; Pivtoraiko et al., 2009; Saxena et al., 2022)) have been developed for motion planning in deterministic environments. Kavraki et al. (1996); LaValle (1998); Kuffner & LaValle (2000) develop sampling-based techniques that randomly sample configurations in the environment and connect them for computing a motion plan from the initial and goal configurations. Holte et al. (1996); Pivtoraiko et al. (2009); Saxena et al. (2022) discretize the configuration space and use search techniques such as A* search to compute motion plans in the discrete space.

Multiple approaches (Du et al., 2010; Kurniawati et al., 2012; Vitus et al., 2012; Berg et al., 2017; Hibbard et al., 2022) have been developed for performing motion planning with stochastic dynamics. Alterovitz et al. (2007) build a weighted graph called stochastic motion roadmap (SMR) inspired from the probabilistic roadmaps (PRM) (Kavraki et al., 1996) where the weights capture the probability of the robot making the corresponding transition. Sun et al. (2016) use linear quadratic regulator -- a linear controller that does not explicitly avoid collisions -- along with value iteration to compute a trajectory that maximizes the expected reward. However, these approaches require an analytical model of the transition probability of the robot's dynamics. Tamar et al. (2016) develop a fully differentiable neural module that approximates the value iteration and can be used for computing solutions for stochastic path planning problems. However, these approaches (Alterovitz et al., 2007; Sun et al., 2016; Tamar et al., 2016) require discretized actions. Du et al. (2010); Van Den Berg et al. (2012) formulate a stochastic motion planning problem as a POMDP to capture the uncertainty in robot sensing and movements. Multiple approaches (Jurgenson & Tamar, 2019; Eysenbach et al., 2019; Jurgenson et al., 2020) design end-to-end reinforcement learning approaches for solving stochastic motion planning problems. These approaches only learn policies to solve one path planning problem at a time and do not transfer knowledge across multiple problems. In contrast, our approach does not require discrete actions and learn options that are transferrable to different problems.

Several approaches have considered the problem of learning task-specific subgoals. Kulkarni et al. (2016); Bacon et al. (2017); Nachum et al. (2018; 2019); Czechowski et al. (2021) use intrinsic reward functions to learn a two-level hierarchical policy. The high-level policy predicts a subgoal that the low-level goal-conditioned policy should achieve. The high-level and low-level policies are then trained simultaneously using simulations in the environment. Paul et al. (2019) combine imitation learning with reinforcement learning for identifying subgoals from expert trajectories and

bootstrap reinforcement learning. Levy et al. (2019) learn a multi-level policy where each level learns subgoals for the policy at the lower level using Hindsight Experience Replay (HER) (Andrychowicz et al., 2017) for control problems rather than long-horizon motion planning problems in deterministic settings. Kim et al. (2021) randomly sample subgoals in the environment and use a path planning algorithm to select the closest subgoal and learn a policy that achieves this subgoal.

Numerous approaches (Stolle & Precup, 2002; Şimşek et al., 2005; Brunskill & Li, 2014; Kurutach et al., 2018; Eysenbach et al., 2019; Bagaria & Konidaris, 2020; Bagaria et al., 2021) perform hierarchical learning by identifying task-specific options through experience collected in the test environment and then use these options (Sutton et al., 1999) along with low-level primitive actions. Stolle & Precup (2002); Şimşek et al. (2005) lay foundation for discovering options in discrete settings. They collect trajectories in the environment and use them to identify high-frequency states in the environment. These states are used as termination sets of the options and initiation sets are derived by selecting states that lead to these high-frequency states. Once options are identified, they use Q-learning to learn policies for these options independently to formulate Semi-MDPs (Sutton et al., 1999). Bagaria & Konidaris (2020) learn options in a reverse fashion. They compute trajectories in the environment that reaches the goal state. In these trajectories, they use the last $K$ points to define an option. They use these points to define the initiation set of the option and the goal state is used as the termination set. They continue to partition rest of collected trajectories similarly and generate a fixed number (a hyperparameter) of options.

Approaches for combining symbolic planning with reinforcement learning (Silver & Ciosek, 2012; Yang et al., 2018; Jinnai et al., 2019; Lyu et al., 2019; Kokel et al., 2021) use pre-defined abstract models to combine symbolic planning with reinforcement learning. In contrast, our approach learns such options (including initiation and termination sets) as well as their policies and uses them to compute solutions for stochastic path planning problems with continuous state and action spaces. Now, we discuss some of the existing concepts required to understand our approach.

## 3 BACKGROUND

**Motion planning**    Let $\mathcal{X} = \mathcal{X}_{\text{free}} \cup \mathcal{X}_{\text{obs}}$ be the configuration space of a robot $R$ and let $O$ be the set of obstacles in a given environment. Given a collision function $f : \mathcal{X} \to \{0, 1\}$, $\mathcal{X}_{\text{free}}$ represents the set of configurations that are not in collision with any obstacle $o \in O$ such that $f(x) = 0$ and $\mathcal{X}_{\text{obs}}$ represents the set of configurations in collision such that $f(x) = 1$. Let $x_i \in \mathcal{X}_{\text{free}}$ be the initial configuration of the robot and $x_g \in \mathcal{X}_{\text{free}}$ be the goal configuration of the robot. The motion planning problem can be defined as:

**Definition 1.** *A **motion planning problem** $\mathcal{M}$ is defined as a 4-tuple $\langle \mathcal{X}, f, x_i, x_g \rangle$, where $\mathcal{X} = \mathcal{X}_{free} \cup \mathcal{X}_{obs}$ is the configuration space, $f$ is the collision function, $x_i$ is an initial configuration, and $x_g$ is a goal configuration.*

A solution to a motion planning problem is a motion plan $\tau$. A motion plan is a sequence of configurations $\langle x_0, \ldots, x_n \rangle$ such that $x_0 = x_i$, $x_n = x_g$, and $\forall x \in \tau, f(x) = 0$. Robots use controller that accepts sequenced configurations from the motion plan and generates controls that take the robot from one configuration to the next configuration. In practice, these environment dynamics are noisy, which introduces stochasticity in the overall motion planning problem. This stochasticity can be handled by computing a motion policy $\pi : \mathcal{X} \to \mathcal{X}$ that takes the current configuration of the robot and generates the next waypoint for the robot.

**Markov decision process**    In this work, we define the stochastic path planning problem as a continuous stochastic shortest path (SSP) problem (Bertsekas & Tsitsiklis, 1991). A continuous stochastic shortest path problem is defined as a 5-tuple $\langle \mathcal{X}, \mathcal{A}, T, C, s_0, G \rangle$ where $\mathcal{X}$ is a continuous state space (configuration space of the robot), $\mathcal{A}$ is a set of continuous actions, $T : \mathcal{X} \times \mathcal{A} \times \mathcal{X} \to [0, 1]$ is a transition function, $C : \mathcal{X} \to \mathbb{R}$ is a cost function, $s_0$ is an initial state, and $G \subseteq \mathcal{S}$ is a set of goal states. Discount factor $\gamma$ is set to 1 for this work. A solution to an SSP is a policy $\pi : \mathcal{X} \to \mathcal{A}$ that maps states to actions that take the robot to the goal states and minimizes the cumulative cost. Dynamic programming methods such as value iteration (VI) or policy iteration (PI) can be used to compute such policies when every component of the MDP is known. When one or more SSP components are unknown various reinforcement learning (RL) approaches (Watkins, 1989; Mnih et al., 2015; Lillicrap et al., 2016; Haarnoja et al., 2018) can be used to learn policies.

**Options framework** Options framework (Sutton et al., 1999) models options as temporal abstraction over primitive actions in an MDP. Let $\mathcal{S}$ be the state-space of an MDP. An option $o$ is defined as 3-tuple $\langle \mathcal{I}_o, \beta_o, \pi_o \rangle$. $\mathcal{I}_o \subseteq \mathcal{S}$ defines the initiation set of an option $o$. An option $o$ can be applied in a state $s$ iff $s \in \mathcal{I}_o$. $\beta_o$ is the termination set of an option $o$. Execution of an option $o$ terminates when the agent reaches a state $s \in \beta_o$. $\pi_o : \mathcal{S} \to \mathcal{A}$ defines policy for the option $o$. Sutton et al. (1999) define Semi-Markov Decision Process (SMDP) by adding available options to the set of primitive actions available in the MDP. Solution for an SMDP is a policy $\pi : \mathcal{S} \to \bar{\mathcal{A}}$ where $\bar{\mathcal{A}} = \mathcal{A} \cup \mathcal{O}$ and $\mathcal{O}$ is a set of options.

**Automated synthesis of abstractions** Shah & Srivastava (2022) use region-based Voronoi diagrams (RBVDs) to define abstractions for motion planning problems. To construct effective abstractions, they use *critical regions* (Molina et al., 2020) to construct the RBVD. Intuitively, critical regions are defined as regions that have a high density of solutions for a given class of problems but low sampling probability under uniform sampling distribution. Shah & Srivastava (2022) use critical regions to define RBVDs as follows:

**Definition 2.** *Given a configuration space $\mathcal{X}$, let $d^c$ define minimum distance between a configuration $x \in \mathcal{X}$ and a region $\phi \subseteq \mathcal{X}$. Given a set of regions $\Phi$ and a robot $R$, a **region-based Voronoi diagram** $\Psi$ is a partitioning of $\mathcal{X}$ such that for every Voronoi cell $\psi_i \in \Psi$ there exists a region $\phi_i \in \Phi$ such that forall $x \in \psi$ and forall $\phi_j \neq \phi_i$, $d^c(x, \phi_i) < d^c(x, \phi_j)$ and each $\psi_i$ is connected.*

In this framework, abstract states are defined using a bijective function $\ell : \Psi \to \mathcal{S}$ that maps each Voronoi cell to an abstract state. The abstraction function $\alpha : \mathcal{X} \to \mathcal{S}$ is defined such that $\alpha(x) = s$ if and only if there exists a Voronoi cell $\psi$ such that $x \in \psi$ and $\ell(\psi) = s$. A configuration $x \in \mathcal{X}$ is said to be in the *high-level abstract state $s \in \mathcal{S}$* (denoted by $x \in s$) if $\alpha(x) = s$. In this work, we use RBVDs along with critical regions to construct abstractions and synthesize options.

## 4 OUR APPROACH

---

**Algorithm 1:** Stochastic Hierarchical Abstraction-guided Robot Planner (SHARP)

---

**Input:** Training environments $E_{\text{train}}$, c-space $\mathcal{X}$, initial configuration $x_i$, goal configuration $x_g$

**Output:** A policy $\Pi$ composed of options

1 $\Theta \leftarrow$ get_critical_regions_predicter();
2 **if** $\Theta$ *is not trained* **then**
3 $\quad$ train $\Theta$ using $E_{\text{train}}$

4 **if** *abstraction is **not** constructed* **then**
5 $\quad$ $\Phi \leftarrow$ predict_critical_regions($e_{\text{test}}, \Theta$);
6 $\quad$ $\Psi \leftarrow$ construct_RBVD($e_{\text{test}}, \Phi$);
7 $\quad$ $\mathcal{O} \leftarrow$ synthesize_option_endpoints($\Phi, \Psi$);

8 $s_i, s_g \leftarrow$ get_abstract_states($x_i, x_g$);
9 $p \leftarrow$ A*_search($\mathcal{O}, s_i, s_g$);
10 $\Pi =$ empty_list;
11 $\pi_0 \leftarrow$ learn_policy($x_i, \mathcal{I}_{o_1}$);
12 $\Pi$.add($\pi_0$);
13 **foreach** $o \in p$ **do**
14 $\quad$ **if** *o.policy does **not** exist* **then**
15 $\quad\quad$ $G \leftarrow$ compute_option_guide($\mathcal{I}_o, \beta_o$);
16 $\quad\quad$ train $o$.policy;
17 $\quad\quad$ adjust the cost of the option $o$;
18 $\quad$ $\Pi$.add($o$.policy);

19 $\pi_{n+1} \leftarrow$ learn_policy($\beta_{o_n}, x_g$);
20 $\Pi$.add($\pi_{n+1}$);
21 **return** $\Pi$;

---

Alg. 1 outlines our approach -- **s**tochastic **h**ierarchical **a**bstraction-guided **r**obot **p**lanner (SHARP) -- for computing a policy for the given stochastic path planning problem. Our approach requires a robot simulator to compute path plans. It takes the occupancy matrix of the environment along with the initial and goal configuration of the robot as the input and produces a policy composed of options as the output.

**Learning the critical regions** The first step in our approach is to identify critical regions in the given environment. We use a deep neural network to identify these critical regions. We train (lines 1-3) this neural network with the kinematic model of the robot and a set of training environments $E_{\text{train}}$ that do not include the test environments. We discuss this in detail in Sec. 4.1. Once the network is trained, the same network is used to identify critical regions (line 5) in all test environments for the same robot.

**Synthesizing option endpoints** Alg. 1 uses these identified critical regions to construct RBVD (line 6) and generate a set of abstract states (line 7). We then use this RBVD to construct a collection of options and synthesize option endpoints for the given environment (line 7). Sec. 4.2 explains this step in detail. These option are synthesized only once per the robot and the environment. They are reused for each subsequent pair of initial and goal pairs. We refer to option endpoints as options for brevity.

**Options as abstract actions** Let $\mathcal{O}$ be the set of synthesized options. Our approach uses these options as high-level abstract actions. It considers the initiation sets of these options as preconditions

for the abstract actions and the termination sets as their effects. We identify the abstract initial and goal state for the robot's initial and goal configuration (line 8) and use the $A^*$ search to compute a sequence of identified abstract actions that takes the robot from the initial abstract state to the goal abstract state (line 9). We use the Euclidean distance between the termination set of the corresponding option and the goal configuration as the heuristic for the $A^*$ search. The $A^*$ search also requires the costs for the options which are initially unknown. Therefore, we use the Euclidean distance between the initiation and termination sets of an option as an approximation to its cost.

**Computing policies for options**    Let $p = \langle o_1, \ldots, o_n \rangle$ be the sequence of options that takes the robot from its initial abstract state $s_0$ to its goal abstract state $s_g$. Alg. 1 then computes policies for each of the options in this sequence of options. It first starts by computing a policy $\pi_0$ that takes the robot from its initial configuration $x_i$ to some point in the initiation set of the first option $o_1$ (line 11). Alg. 1 then starts computing the policy for every option in $p$. First, we check if a policy for an option is already computed from the previous call of Alg. 1 or not (line 14). If it is already computed, we use the same policy. Otherwise, our approach computes an *option guide* for the given option using its endpoints, and it then uses this option guide to compute a policy for the option (lines 15-16). Sec. 4.3 presents this in details. Lastly, Alg. 1 computes a policy that takes the robot from the termination set of the last option in $p$ to the goal configuration $x_g$ (line 19).

**Updating the option costs**    To efficiently transfer the learned option policies, our approach needs to update the option costs that $A^*$ search uses to compute the sequence of options. We update this cost (line 17) by collecting rollouts of the learned policy and using the average number of steps from the initiation set to the termination set as an approximation of the cost of the option. Now, we explain each step of our approach in detail.

## 4.1 LEARNING CRITICAL REGIONS

Our approach first needs to identify critical regions (Sec. 3) to synthesize options in the given environment. Recall that critical regions are regions in the environment that have a high density of solutions for the given class of problems but at the same time, they are hard to sample under uniform distribution over the configuration space. The training data is generated by solving randomly sampled motion planning problems. Input into the network is an occupancy matrix of the environment that represents the free space and obstacles in the given environment. Labels represent critical regions in the given environment.

These critical region predictors are environment independent and they are also generalizable across robots to a large extent. Furthermore, the approach presented here directly used the open-source critical regions predictors made available by Shah & Srivastava (2022). These predictors are environment independent and need to be trained only once per kinematic characteristics of a robot. E.g., the non-holonomic robots used to evaluate our approach (details in Sec. 5) are different from those used by Shah & Srivastava (2022) but we used the critical regions predictor developed by them for a rectangular holonomic robot.

Shah & Srivastava (2022) use 20 training environments ($E_{\text{train}}$) to generate the training data. For each training environment $e_{\text{train}} \in E_{\text{train}}$, they randomly sample 100 goal configurations. Shah & Srivastava (2022) randomly sample 50 initial configurations for each goal configuration and compute motion plans for them using an off-the-shelf motion planner and a kinematic model of the robot. They use UNet (Ronneberger et al., 2015) with *Tanh* activation function for intermediate layers and *Sigmoid* activation for the last layer. They use the weighted logarithmic loss as the loss function. Lastly, they use ADAM optimizer (Kingma & Ba, 2014) with learning rate $10^{-4}$ to minimize the loss function for $50,000$ epochs.

## 4.2 SYNTHESIZING OPTION ENDPOINTS

The central idea of this paper is to synthesize transferable options using the set of critical regions $\Phi$ and the RBVD $\Psi$. Our approach uses these critical regions and RBVD for identifying option endpoints, i.e., initiation and termination sets. In this work, we define two types of options: 1. Options between centroids of two critical regions -- *centroid options* ($\mathcal{O}_c$) and 2. Options between interfaces of two pairs of abstract states -- *interface options* ($\mathcal{O}_i$). Now we define each of these options.

**Centroid options**  Centroid options are defined using centroids of two neighboring critical regions. Intuitively, these options define abstract actions that transition between a pair of critical regions. Formally, let $\Phi$ be the set of critical regions and $\Psi$ be the RBVD constructed using the critical regions $\Phi$. Let $\mathcal{S}_\Psi$ be a set of abstract states defined using the RBVD $\Psi$.

**Definition 3.** *Let $s_i \in \mathcal{S}$ be an abstract state in the RBVD $\Psi$ with the corresponding critical region $\phi_i \in \Phi$. Let $d$ be the Euclidean distance measure and let $t$ define a threshold distance. Let $c_i$ be the centroid of the critical region $r_i$. A **centroid region** of the critical region $r_i$ with the centroid $c_i$ is defined as a set of configuration points $\{x | x \in s_i \land d(x, c_i) < t\}$.*

We use this definition to define the endpoints for the centroid options as follows:

**Definition 4.** *Let $s_i, s_j \in \mathcal{S}$ be a pair of neighboring states in an RBVD $\Psi$ constructed using the set of critical regions $\Phi$. Let $\phi_i, \phi_j \in \Phi$ be the critical regions for the abstract states $s_i$ and $s_j$ and let $c_i$ and $c_j$ be their centroids regions. The **endpoints for a centroid option** are defined as a pair $\langle \mathcal{I}_{ij}, \beta_{ij} \rangle$ such that $\mathcal{I}_{ij} = c_i$ and $\beta_{ij} = c_j$.*

**Interface options**  Interface options are defined using intersections of high-level states. Intuitively, these options define high-level temporally abstracted actions between intersections of pairs of abstract states and take the robot across the high-level states. To construct interface options, we first need to identify interface regions between a pair of neighboring abstract states. Let $\mathcal{S}$ be a set of abstract states defined using the RBVD $\Psi$. We define interface region as follows:

**Definition 5.** *Let $s_i, s_j \in \mathcal{S}$ be a pair of neighboring states and $\phi_i$ and $\phi_j$ be their corresponding critical regions. Let $d^c(x, \phi)$ define the minimum euclidean distance between configuration $x \in \mathcal{X}$ and some point in a region $\phi \subset \mathcal{X}$. Let $p$ be a configuration such that $d^c(p, \phi_i) = d^c(p, \phi_j)$ that is, $p$ is on the border of the Voronoi cells that define $s_i$ and $s_j$. Given the Euclidean distance measure $d$ and a threshold distance $t$, an **interface region** for a pair of neighboring states $(s_i, s_j)$ is defined as a set $\{x | (x \in s_i \lor x \in s_j) \land d(x, p) < t\}$.*

We use this definition of interface regions to define endpoints for the interface options as follows:

**Definition 6.** *Let $s_i, s_j, s_k \in \mathcal{S}_\Psi$ be abstract states in the RBVD $\Psi$ such that $s_i$ and $s_j$ are neighbors and $s_j$ and $s_k$ are neighbors. Let $\hat{\phi}_{ij}$ and $\hat{\phi}_{jk}$ be the interface regions for pairs of high-level states $(s_i, s_j)$ and $(s_j, s_k)$. The **endpoints for an interface option** are defined as a pair $\langle \mathcal{I}_{o_{ijk}}, \beta_{o_{ijk}} \rangle$ such that $\mathcal{I}_{o_{ijk}} = \hat{\phi}_{ij}$ and $\beta_{o_{ijk}} = \hat{\phi}_{jk}$.*

We construct a collection of options for the given environment using these definitions for endpoints of centroid and interface options. Let $\mathcal{V} : \mathcal{S} \times \mathcal{S} \to \{0, 1\}$ define a neighborhood function such that $\mathcal{V}(s_i, s_j) = 1$ iff $s_i, s_j \in \mathcal{S}$ are neighbors. We define the set of centroid options as $\mathcal{O}_c = \{o_{ij} | \forall s_i, s_j \; \mathcal{V}(s_i, s_j) = 1\}$ with endpoints computed as in Def. 4 . Similarly, we define the set of interface options as $\mathcal{O}_i = \{o_{ijk} | \forall s_i, s_j, s_k \; \mathcal{V}(s_i, s_j) = 1 \land \mathcal{V}(s_j, s_k) = 1\}$ with endpoints computed as in Def. 6.

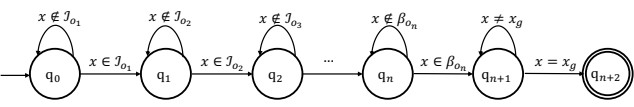

Figure 1: A policy composed of a collection of options $o_1, \ldots, o_n$ for a sequence of distinct adjacent abstract states $s_1, \ldots, s_n$.

Our approach uses this collection of options to compute a composed policy. Let $s_1, \ldots, s_n$ be a set of distinct adjacent states such that $\mathcal{V}(s_i, s_{i+1}) = 1$. Let $\mathcal{O} = \{o_1, \ldots, o_n\}$ be the set of centroid or interface options for these sequence of adjacent states. The **composed policy** $\Pi^{\mathcal{O}}$ is a finite state automaton as shown in Fig. 1. Recall that for a sequence of options $o_1, \ldots, o_n$, Alg. 1 computes $\pi_0$ and $\pi_{n+1}$ as special cases. For a controller state $q_i$, $\Pi(x) = \pi_i(x)$ where $\pi_i$ represents the policy for option $o_i \in \mathcal{O}$. The controller makes a transition $q_i \to q_{i+1}$ when the robot reaches a configuration $x \in \mathcal{I}_{o_{i+1}}$. Now, we prove the existence of a composed policy for our approach. Recall that for a configuration space $\mathcal{X}$ and a set of critical regions $\Phi$, the RBVD $\Psi$ induces a set of abstract state $\mathcal{S}$ and an abstraction function $\alpha$.

**Theorem 4.1.** *Let $x_1$ and $x_n$ be the initial and goal configurations of the robot such that $s_1 = \alpha(x_1)$ and $s_n = \alpha(x_n)$. Let $\mathcal{V} : \mathcal{S} \times \mathcal{S} \to \{0, 1\}$ be the neighborhood function as defined above. Let $\mathcal{O}$ be the set of centroid or interface options. If there exists a sequence of distinct abstract states $s_1, \ldots, s_n$ such that $\mathcal{V}(s_i, s_{i+1}) = 1$ then there exists a composed policy $\Pi$ such that the resultant configuration after the termination of every option in $\Pi$ would be the goal configuration $x_n$.*

*Proof.* (Sketch) The proof directly derives from the definition of the endpoints for the centroid and interface options. Given a sequence of adjacent abstract states $s_1, \ldots, s_n$, Def. 4 and 6 ensures a sequence of options $o_1, \ldots, o_n$ such that $\beta_i = \mathcal{I}_{i+1}$. This implies that an option can be executed once the previous option is terminated. Given this sequence of options $o_1, \ldots, o_n$, according to the definition of the compound policy, there exists a compound policy $\Pi$ such that for every pair of options $o_i, o_j \in \Pi$, $\mathcal{I}_{o_j} = \beta_{o_i}$. Thus, we can say that if every option in $\Pi$ terminates, then the resulting configuration would be the goal configuration.

$\square$

### 4.3 Synthesizing and using Option Guides

Once we identify the endpoints -- initiation and termination sets -- for the option, we use these sets to compute option guides for these options. We use option guides to formulate a reward function that is used to train policies for them. The option guides contain two components: a guide path and a pseudo reward function. We formally define an option guide as follows:

**Definition 7.** *Given an option $o_i \in \mathcal{O}$ with option endpoints $\langle \mathcal{I}_i, \beta_i \rangle$, an **option guide** $G_{o_i}$ is defined as 4-tuple $\langle \mathcal{I}_i, \beta_i, \mathcal{G}_i, R_i \rangle$ where $\mathcal{I}_i$ and $\beta_i$ are endpoints of the option $o_i$, $\mathcal{G}_i$ is the guide path, and $R_i$ is the pseudo reward function.*

We now define both components of the option guide. A guide path is a motion plan from the initiation set to the termination set of an option. Recall that $\alpha$ is an abstraction function defined using the RBVD $\Psi$. We defune a guide path as follows:

**Definition 8.** *Let $\mathcal{X}$ be the configuration space of the robot $R$. Let $d$ be the Euclidean distance measure. Given an option $o_i$ with endpoints $\langle \mathcal{I}_i, \beta_i \rangle$, let $c_{\mathcal{I}_i}$ and $c_{\beta_i}$ define centroids for the initiation and termination sets respectively. Given a threshold distance $t$ and the Euclidean distance measure $d$, a **guide path** $\mathcal{G}_i$ for the option $o_i$ is a sequence of points $[p_1, \ldots, p_n]$ in the configurations space $\mathcal{X}$ such that $p_1 = c_{\mathcal{I}_i}$, $p_n = c_{\beta_i}$, for each pair of consecutive points $p_i, p_j \in \mathcal{G}_i$, $d(p_i, p_j) < t$, and for every point $p_i \in \mathcal{G}_i$, $p_i \in \alpha(\mathcal{I}_i)$ or $p_i \in \alpha(\beta_i)$.*

Here, we abuse the notation and use the abstraction function with a set of low-level configurations rather than a single configuration. Intuitively, $\alpha(\mathcal{I}_i) = \{\alpha(x) | \forall x \in \mathcal{I}_i\}$. It can be similarly computed for $\beta_i$. We generate this guide path using a sampling-based motion planner HARP (Shah & Srivastava, 2022) and the kinematic model of the robot.

The guide path is used to define a pseudo reward function $R$ for the option. It is a dense reward function that provides a reward to the robot according to the distance of the robot from the nearest point in the guide path and the distance from the last point of the guide path. For an option $o_i$, we define the dense pseudo reward function $R_i : \mathcal{X} \to \mathbb{R}$ as follows:

**Definition 9.** *Let $o_i$ be an option with endpoints $\langle \mathcal{I}_i, \beta_i \rangle$ and let $\mathcal{G}_i = [p_1, \ldots, p_n]$ be the guide path. Given a configuration $x \in \mathcal{X}$, let $n(x) = p_i$ define the closest point in the guide path. Let $d$ be the Euclidean distance measure. The **pseudo reward function** $R_i(x)$ is defined as:*

$$R_i(x) = \begin{cases} r_t & \text{if } x \in \beta_i \\ r_p & \text{if } x \in \mathcal{S}/\{\alpha(\mathcal{I}_i), \alpha(\beta_i)\} \\ -(d(x, n(x)) + d(n(x), p_n)) & \text{otherwise} \end{cases}$$

Here $r_t$ is a large positive reward and $r_p$ is a large negative reward that can be tuned as hyperparameters. Intuitively, we wish to automatically generate a dense pseudo-reward function based on the information available form the environment. Thus rather than providing only a sparse reward function (e.g. "+1 when you reach the termination set of the option"), we develop the pseudo reward function that captures this condition (first case), a penalty for straying away from the source and target abstract states (second case), and a reward for covering more of the option guide (third case).

Given an option $o_i$ with endpoints $\langle \mathcal{I}_i, \beta_i \rangle$ and an option guide $G_i = \langle \mathcal{I}_i, \beta_i, \mathcal{G}_i, R_i \rangle$, our approach can use arbitrary reinforcement learning algorithm to learn the policy for the option. For our empirical evaluation (Sec. 5), we use Soft Actor Critic (Haarnoja et al., 2018).

## 5 Empirical Evaluation

We conduct an extensive evaluation of our approach in a total of four different environments with two different robots. All experiments were conducted on a system running Ubuntu 18.04 with AMD

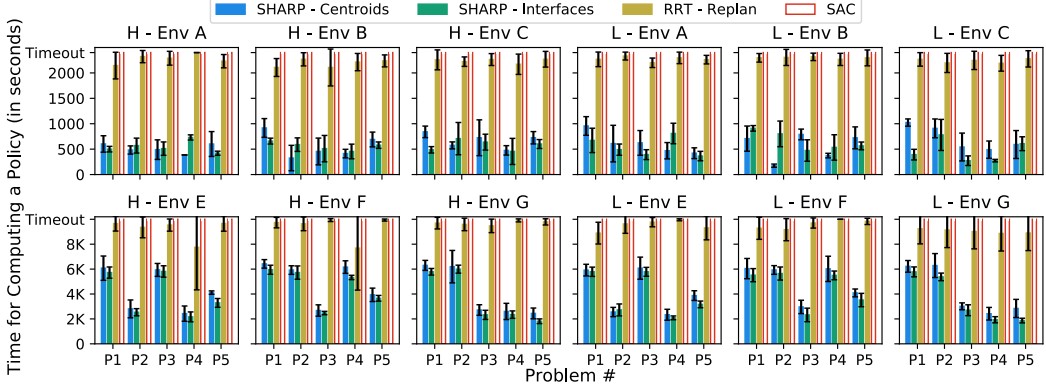

Figure 2: The figure shows the time taken by our approach and baselines to compute path plans in the test environment. Results for an additional environment is in the Appendix B. The x-axis shows the problem instance and the y-axis shows the time in seconds. The reported time for our approach includes time to predict critical regions, construct abstractions, and learn policies for all the options. Each subsequent problem instance uses trained policies for options from the previous problems if there exists one. Timeout was set to 2400 seconds. The numbers are averaged over 5 independent trials. The transparent bars for SAC show that training was stopped as it reached the timeout.

Ryzen Threadripper $3960X$ 24-core processor and two NVidia 3080 GPUs. We implement our approach using PyBullet (Coumans & Bai, 2016) robotics simulator. PyBullet does not explicitly model stochasticity in the movement of the robot. Therefore, we use random perturbations in the actions to introduce stochasticity in the environment while training and use default controllers to evaluate the learned policies.

We implement our approach using PyTorch and Stable Baselines. We use a $2-$layered neural network with each layer containing 256 neurons in every layer to learn policy for each option. The input to the network is the current configuration of the robot and a vector to the nearest point on the guide path for the current option. The network outputs the next point for the robot to reach. We use $r_t = 1000$ and $r_p = -100$ for the pseudo reward function with maximum learning steps set to $150k$ to learn policy for each option. We evaluate our model for 20 episode every $10k$ steps and stop the training if we achieve an average reward of $500$. Our code and data are available in the supplementary material.

**Evaluating our approach** We evaluate our approach using a total of 7 environments. They are inspired by indoor household and office environments. Appendix A shows our test environments. Dimensions of the first four environments (App. A.1) are $15m \times 15m$. The rest of the environments (App. A.2) are of the size $75m \times 75m$. These environments were not included in the set of training environments used to train the network that identifies critical regions. We generate 5 motion planning tasks (P1-P5) for each environment to evaluate our approach. We use two non-holonomic robots to evaluate our approach in these four environments: the ClearPath Husky robot (H) and the AgileX Limo (L). The Husky is a 4-wheeled differential drive robot that can move in one direction and rotate in place. The Limo is also a 4-wheeled robot that can move in one direction and has a steering.

We considered a set of recent approaches (Kulkarni et al., 2016; Lillicrap et al., 2016; Levy et al., 2019; Bagaria & Konidaris, 2020) before selecting two approaches to evaluate against our approach. First, we compare our approach against vanilla soft actor-critic (SAC) (Haarnoja et al., 2018). SAC is an off-policy deep reinforcement learning approach and learns a single policy for taking the robot from its initial configuration to the goal configuration. We use the same network architecture as ours for the SAC neural network. We use the terminal reward of $+1000$ and a step reward of $-1$. We also evaluate our approach against a sampling-based motion planner RRT (LaValle, 1998) with replanning. We compute a path plan with RRT and execute it. If the robot does not reach the goal after executing the entire path plan, then we compute a new path plan from the configuration where the robot ended. We continue this loop until either the robot reaches the goal configuration or we reach the timeout.

**Analysis of the results** We thoroughly evaluated our approach to answer three critical questions: (1) Does this approach learn useful high-level planning representations? (2) Do these learned representations support transferring learning to new tasks?

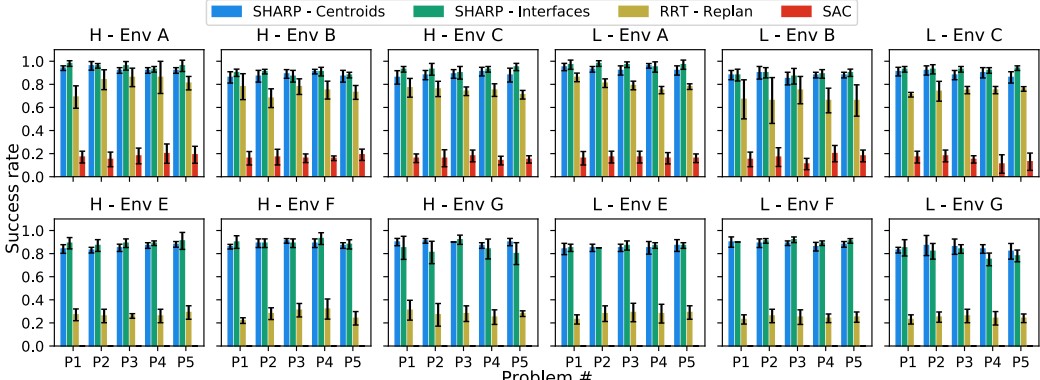

Figure 3: The figure shows the success rate of our approach and baselines in the test environments. Results for an additional environment is in the Appendix B. The x-axis shows the problem instance and the y-axis shows the fraction of successful executions of the model out of 20 test executions. We used the final policy for our approach and SAC. RRT computed a new plan for each execution with a timeout set to 2400 seconds. The numbers are averaged over 5 independent trails.

Fig. 2 shows the time taken by our approach for learning policies compared to the baselines SAC and RRT. Fig. 2 shows that our approach was able to learn policies significantly faster than the baselines. In most cases, our approach was able to compute solutions in half of the time taken by the baselines. Vanilla SAC was not able to reach the threshold average reward of $+500$ in any environment before the timeout of 2400 seconds. This shows that our approach was able to learn useful high-level planning representations that improve the efficiency of the overall stochastic motion planning.

Fig. 3 shows the success rate for our approach and the baselines. Our approach achieves the goal a significantly higher number of times compared to replanning-based deterministic RRT. Our approach was able to consistently reach the goal configuration more than $80\%$ of times. On the other hand, RRT was able to reach the goal only $50\%$ of times in the given timeout of 2400 seconds. For larger environments (E-F), SAC was not able to solve a single problem. We also conducted a thorough analysis of options being used across problems P1-P5 in each environment. Appendix C shows the fraction of options reused by our approach across P1-P5. These reuse rates combined with the success rates show that our approaches is able to successfully transfer learned options across new tasks.

## 6 CONCLUSION

In this paper, we presented the first approach that uses a data-driven process to automatically identify transferable options. Our approach synthesizes two different types of options: centroid options and interface options. It uses these options with our hierarchical algorithm for solving stochastic path planning problems. We also provide a way to automatically generate pseudo reward functions for synthesized options to efficiently learn their policies. We show that learned options can be transferred to different problems and used to solve new problems. Our formal framework also provides guarantees on the compossibility of the generated options.

Our empirical evaluation on a large variety of problem settings shows that our approach significantly outperforms other approaches. Through our empirical evaluation, we show that by combining reinforcement learning with abstractions and high-level planning we can improve its efficiency and compute solutions faster. We also show that using reinforcement learning to solve path planning problems in stochastic environments yields robust plans that perform better than deterministic plans computed using replanning-based motion planners.

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
