# OpenReview forum: "Multi-Task Option Learning and Discovery for Stochastic Path Planning"
_ICLR.cc/2023/Conference — Submitted to ICLR 2023_

### Official Review · Reviewer_CLF4 · 2022-10-22

**Confidence:** 2
**Correctness:** 3
**Technical Novelty And Significance:** 3
**Empirical Novelty And Significance:** 2
**Recommendation:** 5

**Clarity, Quality, Novelty And Reproducibility:**

The paper is mostly clear, but the description of the experimental setup, and its discussion could be improved. At evaluation time, it is not clear to me what the input observations are to the algorithms. Is it the x-y coordinates of the robot? Is it a top down image of the environment? Are the inputs the same across all the algorithms tested?

Further, the comparisons between the algorithms’ performances are made in terms of time. Since they are evaluated in a simulated environment, how do they compare in terms of the number of timesteps or environmental interactions? How do the algorithms compare in terms of the amount of pre-training? This should be discussed.

The work in the paper appears to be novel, and given the details provided I suspect that the results should be reproducible.


**Strength And Weaknesses:**

The main strength of this paper is that it provides a new and complete algorithm for stochastic path planning. It combines conventional path planning with RL, and its effectiveness is supported by experimental results in some limited simulated domains.

The main weaknesses of the paper are the complexity of the proposed algorithm and the limited experimental evaluation. The proposed approach requires a complex series of steps to produce the final policy, and it is not clear to me that this complexity is fully justified. For example, a step in the algorithm is to generate a "guide path", and this is done using the HARP motion planner. Since you are running HARP, why not just use the path planned by HARP to make the policy directly? What is the advantage of turning the path into a reward function, then training a policy on that?

What is not clear from the experimental evaluation is how the method will scale to larger navigation tasks. There is no discussion of this. Will this method be useful in real world robotics navigation problems? It is complicated and requires accurate models: will it scale?


**Summary Of The Paper:**

This paper proposes a new algorithm to learn policies for stochastic path problems that is called stochastic hierarchical abstraction-guided robot planner (SHARP). The algorithm consists of four parts: identify critical regions, synthesize option endpoints, generate a pseudo-reward function, and finally learn an option policy. The new approach is claimed to be able to transfer knowledge across tasks, and outperform existing approaches. The claims are supported by experiments in simulated robot navigation domains.

**Summary Of The Review:**

Even though the proposed algorithm appears to be novel and interesting, the experimental evaluation is limited. For this reason I suggest rejecting this paper.

The paper could be improved by some additional experiments on larger and/or more complicated robot navigation tasks. Even including a discussion of how SHARP would scale to larger domains, or to the real world, would be a significant improvement.

---

### Official Review · Reviewer_Xjkg · 2022-10-23

**Confidence:** 4
**Correctness:** 3
**Technical Novelty And Significance:** 2
**Empirical Novelty And Significance:** 2
**Recommendation:** 3

**Clarity, Quality, Novelty And Reproducibility:**

The paper is not crisp (see below). The amount of novelty in the paper seems marginal. Authors discussed about sharing code in supplementary material, but I could not find them.

Readability:
- Until the experimentation section, the paper gives the impression that centroid and interface options are gonna be used jointly which causes a lot of confusion. Later in experimentation the reader sees two set of results. Please clarify this early on.
- I encouraged to use a different notation for region for better readability as most RL folks will view r as the reward.
- "only once per the robot and the environment"
- why use n as a function given that was used earlier as subscript index?
- The white color for SAC does not provide good visibility in figure 3. Please use the same color as used in figure 4. Also why SAC does not have error bar in Figure 3?
- Figure 2: Would be very helpful to also show one 3D snapshot of the environment.
- Several citations have redundant years, example:
"Naman Shah and Siddharth Srivastava. Using deep learning to bootstrap abstractions for hierarchical robot planning. In Proc. AAMAS, 2022, 2022."

Questions:
- A* requires an admissible heuristic. If you are using the average roll-out value for your options to update heuristics, wouldn't that break this assumption for the high-level planning phase?


**Strength And Weaknesses:**

Strengths
+ Option discovery and planning using abstraction for stochastic path planning is of huge interest to the community.
+ The main idea of the paper is simple.

Weaknesses
- Writing: The paper is hard to follow (see below)
- The literature review misses key references (e.g. [1]).
- The baseline for empirical results is weak. Two baseline to include [2] using hard-coded options and [1]
- "Does combining reinforcement learning with high-level planning improve its efficiency?", the answer is yes and already covered in [2]

[1] Pierre-Luc Bacon, Jean Harb, Doina Precup, "The Option-Critic Architecture", in Thirthy-first AAAI Conference On Artificial Intelligence 2017
[2] Richard S Sutton, Doina Precup, and Satinder Singh. "Between mdps and semi-mdps: A framework for temporal abstraction in reinforcement learnings". Artificial intelligence, 112(1-2):181–211, 1999.



**Summary Of The Paper:**

Authors tackled the problem of stochastic path planning using option learning. The idea is to a) sample good trajectories to identify critical regions based on region-based Voronoi diagrams [shah & Srivastava 2022] b) define options based on critical regions, authors explored both centroid based and interface based c) learn the corresponding policies for discovered options d) find a path on the abstracted space using A* e) project the high-level path to lower-level actions using the option policies. The approach was later compared against stochastic actor-critic (SAC) and Rapidly-Exploring Random Tree (RRT) showing advantage in terms of planning time and path success rate.


**Summary Of The Review:**

Love the direction of the paper, but it needs to be improved in terms of writing and positioning with existing efforts to be fully baked.

---

### Official Review · Reviewer_Dfkf · 2022-10-24

**Confidence:** 4
**Correctness:** 3
**Technical Novelty And Significance:** 2
**Empirical Novelty And Significance:** 2
**Recommendation:** 6

**Clarity, Quality, Novelty And Reproducibility:**

The writing is clear and easily understood. The proposed algorithm is original in combining option-based high-level planning with reinforcement learning policies for stochastic path planning. The clarity of the experimental analysis section needs to be improved and I have listed some of the current shortcomings above.

**Strength And Weaknesses:**

The paper contains a concise overview of the related works and the background concepts necessary to understand the proposed approach to stochastic path planning.

The following points should be addressed to improve the paper:

1. In Definition 9, the reward function formulation lacks intuition / motivation. It is not clear to me why this particular formulation was proposed and what is the difference between the second and third case for the definition of $R_i(x)$.

2. The description of the experimental setup is confusing to me.

- (a) I think P1-P5 are the 5 start and goal tasks but it has not been explicitly mentioned in the paper.

- (b) The results show that SHARP-Centroids can sometimes plan faster than SHARP-Interfaces, and vice versa. What if any is the key difference between the two types of options? Is it task dependent or is there a general characterization that can differentiate the performance of the two approaches?

- (c) “Fig 3 and 4 show that our approach was able to successfully transfer learned policies for options for subsequent problem instances without having to compute these policies again” - This is unclear to me. Some more intuition or explanation for transferability among options would be helpful. Are option policies transferable if the option endpoints are the same across different environments or across different start and goal states in the same environment?


**Summary Of The Paper:**

This paper proposes a stochastic path planning algorithm using options to learn a policy that outperforms only RL-based (SAC) or replanning-based (RRT) motion planners. The proposed algorithm, called SHARP, proceeds hierarchically in 3 steps: (1) The search space is segmented into critical regions using a trained UNet. (2) A sequence of option start and end states are constructed using the critical regions, the start and goal locations, and a sequence of intermediate option guides are synthesized. (3) The options serve as the abstractions over the primitive action space. A* search is used to first plan a path over options. Finally, SAC is used to learn an RL policy that guides the agent to select primitive actions for each option.


**Summary Of The Review:**

Some more intuitions and motivation for the proposed approach, and better analysis of the experimental results would help improve the paper.


==============

Post-rebuttal update: I will keep my original rating of the paper. I agree with reviewer CLF4 regarding insufficient motivation for the real-world applicability of this method in terms of its computational complexity. Although during the rebuttal the authors have provided results on a larger environment (75x75m), Fig 2 also shows that SHARP requires ~6k seconds to find the goal in certain tasks (P1, P2). It would help to provide results on more tasks (currently only 5 tasks are considered), and a thorough analysis of what kinds of tasks SHARP provides an advantage on versus where it can struggle to find the goal and require more time?

---

### Official Review · Reviewer_pCv5 · 2022-10-28

**Confidence:** 3
**Correctness:** 4
**Technical Novelty And Significance:** 3
**Empirical Novelty And Significance:** 3
**Recommendation:** 6

**Clarity, Quality, Novelty And Reproducibility:**

The paper is clear. I really like Algorithm 1, for clarity. With source code in the appendix (later a link on github?) then the work should be reproducible, but I didn't test the code. From an RL perspective, the ideas and the approach is novel.

**Strength And Weaknesses:**

Strength:
* Abstract actions address one of the brittleness-issues of contemporary DRL. Generalization to new maps is demonstrated, which is a major contribution of the paper.
* Abstract actions makes the approach more transparent and interpretable - important properties from the perspective of Trustworthy AI. An abstract plan is much easier to understand, explain and monitor than a DRL policy.
* The approach present an integrated solution, from critical region learning & detection to the hierarchical algorithm that use abstract actions to fine motion plans. It necessitates that the individual parts works in a real way, such that the integrated whole works.
* Connections to graph search and symbolic action/plan representations (reminds me a bit about landmarks or similar from the automatic planning field) allow the method to use and combine the strengths of different fields (A* and RL, symbols and state-space etc.)

Weaknesses:
* \mathcal{S} is used for both state space (configuration space) and for abstract states? Since the state space \mathcal{S} is equal to \mathcal{X}, why not use \mathcal{X} instead?
* "This shows that using RL for stochastic motion planning produces robust solutions.": It is shown to be more robust than RRT (in these environments), but 90% success rate is not *robust*.
* Not necessarily something that has to change, but there are possibly more suitable baselines to compare with from the path/motion planning literature. Large-scale path planning has and still is often approached using hierarchical planning, e.g. by A* over an "abstract" graph, then A* in between the nodes of the graph [1]. Modern non-learning based motion planning approaches (e.g. lattice-based motion planning [2]) use translation invariant motion primitives (generated offline using optimal control) as *actions* in e.g. A*-search to generate physically feasible trajectories in large and complex environments in real-time, with both static and dynamic obstacles. Such methods would provide probably provide a more suitable baseline than RRT (with or without hierarchical comparisons.)
* Why call it "Multi-Task" when it is only path planning? I get that it is for many possible different path goals.


[1] Holte, Robert C., et al. "Hierarchical A*: Searching abstraction hierarchies efficiently." AAAI/IAAI, Vol. 1. 1996.

[2] Pivtoraiko, Mihail, Ross A. Knepper, and Alonzo Kelly. "Differentially constrained mobile robot motion planning in state lattices." Journal of Field Robotics 26.3 (2009): 308-333.

**Summary Of The Paper:**

The authors propose to, for a specific robot, learn to recognize possible intermediary goals for motion tasks which generalize to new previously unseen maps with static obstacles. These possible intermediary motion goals (options) are either centroids (one per region) and interfaces (one per pair of connected regions). Abstract actions are formed by the transition between two options. RL policys are learned for motion of abstract actions, and shown in the evaluation to generalize to new maps. Motion planning takes place first at the abstract action level, and the plan is realized by following the policy of each abstract action, in sequence, until the robot reach the goal. The approach is compared to, and shown to outperform, SAC and RRT in both computation time and motion goal success rate (within a timelimit of 1200 seconds).

**Summary Of The Review:**

The proposed approach is interesting, well-integrated and show benefits over RL without abstract actions for motion planning.
Ideas presented are in part explored in the literature in related fields. Since these are important for the current work it might be good to highlight the connections.

---

### Decision · Program_Chairs · 2023-01-20

**Decision:**

Reject

**Justification For Why Not Higher Score:**

None of the reviewers found the paper strongly convincing and there were significant negative aspects pointed out.

It looks attractive to reviewers mostly familiar with reinforcement learning, because this strategy does work better than HRL.  But the paper does not convince the reader that its method works better than any number of sensible strategies from the robotics literature.

Furthermore, the weak handling of stochasticity, when it is featured in the title, makes the contribution even less clear.

**Justification For Why Not Lower Score:**

N/A

**Metareview: Summary, Strengths And Weaknesses:**

This paper provides a strategy for constructing a set of options and high-level option policies for robot navigation in continuous stochastic domains.  It uses a neural network to map the domain's occupancy grid into a set of "critical regions" which are parts of the workspace that are frequently involved in solution paths but which would be difficult to sample.  These regions are used to define a partition of the space, and then options are constructed for driving between these regions.   The options span the space in the sense that a high-level policy can be constructed out of them to solve any new problem.

The overall idea, of finding a good hierarchical decomposition of the space and using that to define options is a good one.  That idea is novel, from the perspective of current hierarchical RL practice, but as one reviewer points out, in robotics there has been a great deal of work on hierarchical approaches to path finding.

It is true that, as the authors point out, this paper is addressing stochastic problems (unlike the standard robotics hierarchical pathfinding work).  However, the paper fall substantially short in its clarity about exactly how the problem is modeled:
- are the critical regions, abstract states, and option targets in configuration space or workspace?
- what is the model of stochasticity?  do you have some notion of local controllability that means the robot (is likely whp) to stay within one region before traversing to the next?  if not, how do you model the probability the robot might end up in a non-target region? (There are lots of papers in the robotics literature about path finding in stochastic domains that really seriously model the stochasticity (e.g., Huynh, Karaman & Frazzoli).
- if the dynamics are stochastic, how do the control policies 'hedge their bets' to avoid, e.g., going too close to an obstacle that they might crash into?

A critical avenue for improvement of this paper is to adopt a concrete problem framing that might occur for a real robot (including model of stochasticity, etc.) and be sure to make the best baselines you can.  For example, with a nearly holonomic robot and low levels of noise, it might be good to try an existing hierarchical decomposition method for deterministic domains, but make good local controllers that can provably reject some amount of local stochastic perturbation from the environment.